# Midkine Is a Potential Therapeutic Target of Tumorigenesis, Angiogenesis, and Metastasis in Non-Small Cell Lung Cancer

**DOI:** 10.3390/cancers12092402

**Published:** 2020-08-24

**Authors:** Dong Hoon Shin, Jeong Yeon Jo, Sun Ha Kim, Minyoung Choi, Chungyong Han, Beom K. Choi, Sang Soo Kim

**Affiliations:** 1Research Institute, National Cancer Center, Goyang-si, Gyeonggi-do 10408, Korea; 1707101@ncc.re.kr (J.Y.J.); 1905205@ncc.re.kr (S.H.K.); cmy413@ncc.re.kr (M.C.); chungyong.han@ncc.re.kr (C.H.); 11380@ncc.re.kr (B.K.C.); sangsookim@ncc.re.kr (S.S.K.); 2Cancer Biomedical Science, Graduate School of Cancer Science and Policy, National Cancer Center, Goyang-si, Gyeonggi-do 10408, Korea

**Keywords:** midkine, tumor progression, metastasis, angiogenesi, NSCLC

## Abstract

Hypoxia-inducible factors (HIFs) induced by reduced O_2_ availability activate the transcription of target genes encoding proteins that play important roles in communication between cancer and stromal cells. Cancer cells were incubated under hypoxic conditions: H1299, A549 (NSCLC); Hep3B, HepG2 (HCC); HCT116, CT26 (Colon cancer); MCF-7, MDAMB231 (Breast cancer); MKN1, MKN5 (Gastric cancer); U87MG, SHSY5Y (Brain cancer); and SKOV3, SNU840 (Ovary cancer). All cells expressed HIF-1α and HIF-2α mRNA and proteins. However, cell proliferation of NSCLC, breast, gastric, and brain cancer cells under hypoxia was more dependent on HIF-1α except for HCC cells where it was more dependent on HIF-2α. Among HIF-1α dependent cells H1299 was the most affected in terms of cell proliferation by HIF-1α knockdown. To examine which cytokines are secreted in NSCLC cells by HIF-1α to communicate with stromal cells, we performed a cytokine-profiling array with H1299. We screened the top 14 cytokines which were dependent on the HIF-1α expression pattern. Among them, midkine (MDK) expression was affected the most in response to HIF-1α. MDK is a heparin-binding growth factor that promotes angiogenesis and carcinogenesis. Indeed, MDK significantly increased HUVEV endothelial cell migration and neo- vascularization in chick chorioallantoic membrane assay (CAM) assay via paracrine signaling. In addition, MDK secreted from NSCLC cells interacted with Notch2 which activated the Notch signaling pathway and induced EMT, upregulated NF-κB, and increased cancer promotion. However, in response to MDK knock down, siRNA or the MDK inhibitor, iMDK treatment not only decreased MDK-induced migration and angiogenesis of endothelial cells but also abrogated the progression and metastasis of NSCLC cells in in vitro and in vivo orthotopic and spontaneous lung metastasis models. Consequently, iMDK treatment significantly increased mice survival rates compared with the control or MDK expression group. MDK plays a very important role in the progression and metastasis of NSCLC cells. Moreover, the MDK targeting strategy provides a potential therapeutic target for the treatment of MDK-expressing lung cancers.

## 1. Introduction

The tumor microenvironment differs from that of the normal tissue environment [1,2]. The critical difference between the tumor microenvironment and that of the surrounding normal tissue is the presence of intratumoral hypoxia [3]. In cancers such as cervical cancer, soft tissue sarcoma, and breast cancer, where tumor oxygenation is directly measured, hypoxic conditions (pO_2_ < 10 mmHg) prevail in the affected cells when compared with normal tissues (pO_2_ > 65 mmHg) [4,5,6]. Reduced O_2_ availability in hypoxia is a stimulus for the activation of hypoxia-inducible factor 1 (HIF-1), which is a heterodimeric protein composed of HIF-1α and HIF-1β. In addition, there are three HIF family members, HIF-1, HIF-2, and HIF-3 [7,8]. During hypoxia, proteasomal degradation of the complex containing HIF-1α and HIF-1β is inhibited, and the complex is transported to the nucleus, where it binds to HIF response elements (HREs, 5′-RCGTG-3′). The binding of the HIF complex to HREs triggers the recruitment of coactivator molecules that form transcription initiation complexes to enhance the expression of genes that mediate cellular and physiological responses to hypoxia [9]. Even though a number of genes are activated similarly by both HIF-1α and HIF-2α, glycolysis-related genes such as *PGK1* and *LDHA* are up-regulated by HIF-1α, and some genes such as *CITED2*, *PKIB* and *GADD45B* are upregulated more than 2-fold by HIF-2α but not by HIF-1α [10]. In this study, we found that midkine (MDK) is up-regulated by hypoxia, more specifically, MDK mRNA, and the expression and secretion of the protein are regulated by HIF-1α in lung cancer cells.

MDK is a heparin-binding growth factor first discovered as a highly expressed gene during mouse embryogenesis [11]. MDK is viewed as a multi-functional protein and along with pleiotrophin (PTN), they form a structurally unique family of heparin-binding growth factors [12]. Of note, the tumor growth-promoting activity of MDK is also partially due to its ability to promote tumor angiogenesis as a potent proangiogenic factor [13,14]. In addition, MDK also has been linked to cancer cell invasion and metastasis via epithelial-mesenchymal transition (EMT) which has three major pathways: WNT signaling, TGF-β signaling, and Notch2 signaling pathways [15,16]. In particular, the interactions between Notch2 receptor and MDK in pancreatic ductal adenocarcinoma cells (PDACs) and HecaT cells activate the Notch signaling mechanism which is linked to the upregulation of Notch downstream signaling molecules, such as NF-κB and Hes-1 [15,17,18].

MDK is part of the six-biomarker blood test which was developed for the detection of early stage lung cancer in at-risk populations [19]. MDK mRNA and protein overexpression correlated with malignant status and poor prognosis in NSCLC patients [20]. Hao et al. screened the MDK inhibitor, iMDK, which inhibited MDK-positive H441 and H520 lung adenocarcinoma cells [21]. In other study, Masui M. et al. also studied the antitumor effect of iMDK against HSC-2 and SAS cell oral squamous cell carcinoma [22]. In the present study, mRNA and protein levels of MDK were found to be overexpressed in NSCLC cells under hypoxia. Secreted MDK elevated angiogenesis by increasing the interactions of endothelial cells and lung cancer cells via paracrine signaling. Moreover, MDK increased the EMT ability and proliferation pathway of lung cancer cells via the autocrine pathway. However, MDK siRNA or iMDK, which is an MDK inhibitor, decreased endothelial cell migration, tumor promotion, and metastasis both in vitro and in vivo. Targeting the expression of MDK provides a new therapeutic approach for the treatment of MDK-expressing NSCLCs. 

## 2. Materials and Methods

### 2.1. Reagents and Antibodies

HIF-1α plasmid (SC119189) and MDK plasmid (SC319913) were purchased from OriGene Technologies, Inc. (Rockville, MD, USA). Small interfering (si)RNAS, siHIF-1α (sc-35561), siHIF-2α (sc-35316), and siMDK (sc-39711) were purchased from Santa Cruz Biotechnology, Inc. (Dallas, TX, USA). Antibodies against HIF-1α (14179), HIF-2α (59973), β-Actin (3700), phospho-PI3K (17336), PI3K (4249), phospho-Akt (4060), Akt (2920), Notch2 (5732), NF-κB (8242), Hes-1 (11988), E-cadherin (14472), Vimentin (5741), Snail (3879), Survivin (2808), XIAP (14334), BAD (9268), and CD31 (77699) were obtained from Cell signaling Technology, Inc. (Danvers, MA, USA). Anti-MDK antibody (AF-258) from R&D systems (Abingdon, OX, UK) and anti-SV40T Ag antibody (sc-147) from Santa Cruz Biotechnology, Inc., were also purchased. Additionally, the rhVEGF 165 protein (293-VE, R&D systems, Abingdon, OX, UK) and rhMDK (450-16, Peptptech, RockyHill, NJ, USA) were also purchased. iMDK (5126) was purchased from R&D systems (Abingdon, OX, UK).

### 2.2. Cell Lines and Culture Conditions

H1299, A549 (NSCLC); Hep3B, HepG2 (HCC); HCT116, CT26 (Colon cancer); MCF-7, MDAMB231 (Breast cancer); MKN1, MKN5 (Gastric cancer); U87MG, SHSY5Y (Brain cancer); SKOV3, SNU840 (Ovary cancer), BEAS-2B cell line (human normal bronchial epithelial), CCD-18Lu (human fibroblasts); and THP-1 (human monocytic cells) were obtained from the American Type Culture Collection (Manassas, VA, USA). HUVEC (human umbilical vein endothelial cells) were obtained from Lonza (Walkersville, MD, USA). All the cells were maintained in minimum essential medium RPMI1640, EMEM (Hyclone, Logan, UT, USA) or medium 199 (Sigma–Aldrich, St Louis, MO, USA) supplemented with 10% heat-inactivated fetal bovine serum (Hyclone) and antibiotics (Gibco, Grand Island, NY, USA). Cells were incubated in a humidified atmosphere at 37 °C at 20% O_2_/5% CO_2_ for normoxic conditions or at 1% O_2_/5% CO_2_ for hypoxic conditions. All the cell lines were authenticated and were free of mycoplasma contamination. 

### 2.3. Midkine ELISA Assay

The media were collected, stored as frozen at −80 °C, and then thawed on ice prior to the analyses. An enzyme-linked immunosorbent assay (ELISA) kit for human midkine (BioRay Inc., Norcross, WA, USA) was used to measure midkine concentrations in media from cells cultured under normoxic or hypoxic conditions. Midkine levels were determined by following the manufacturer’s protocol. Briefly, 100 μL samples were added to each well and incubated for 2.5 h at room temperature. After incubation, the plates were washed and then biotinylated antibody was added. Streptavidin solution was added to each well and incubated. TMB One-Step Substrate Reagent was added, and the reaction stopped by stop solution. Optical density (OD) was measured at 450 nm in a micro-plate reader (VERSAmax, San Jose, CA, USA). 

### 2.4. Cell Migration Assay

The migration assay was performed using an 8.0 μm pore size transwell membrane chamber. Each transwell was coated with collagen (Merck Millipore (08-115), Frankfurter, Germany) (0.5 mg/mL, outside). Cells (5 × 10^4^ cells) in serum-free RPMI, EMEM, or M199 medium were loaded in the top chamber with BSA, and media from cells cultured under normoxic/hypoxic conditions were loaded in the bottom chamber as a chemoattractant. After incubation at 37 °C for 16 h, the cells on the top of the filters were removed with cotton tips. The cells on the lower surface of the filters were fixed in methanol and stained with 0.1% crystal violet.

### 2.5. Collection of Conditioned Media (CM)

The transfected cells were allowed to stabilize for 24 h, and the cells were seeded at 5 × 10^4^ cells/mL in 100 mm culture dishes with serum-free media without antibiotics prior to incubation in normoxic and hypoxic chambers for 24 h. CM were collected and centrifuged to remove any residual cells and were filtered using a 0.22 μm pore size syringe filter. Filtered CM were concentrated 10-fold using Amicon Ultra-15 (Millipore, Bedford, MA, USA) centrifugal filters at 4 °C. After collecting CM, the number of cells in the dish was counted and normalized to the volume of CM used in each experiment.

### 2.6. Cytokine Profiling Antibody Array

Proteins from CM were obtained using a gel matrix column that was included in the antibody array assay kit (Full Moon Biosystems, Sunnyvale, CA, USA). After hydration, the column was centrifuged and the protein concentration measured using the bicinchoninic acid assay (BCA) protein assay kit (Pierce, Rockford, IL, USA). For each sample, 75 μL of labeling buffer was added to 60 μg protein sample. Subsequently, the sample was treated with biotin/DMF solution. The cytokine-profiling antibody array slide (Full Moon Biosystems, Sunnyvale, CA, USA) was treated with 30 mL of blocking solution in a Petri dish and incubated for 1 h at RT with shaking. The blocked array slide was incubated with coupling mixture. Cy3-streptavidin (GE Healthcare, Chalfont St. Giles, UK) was mixed with 30 mL of detection buffer. The coupled array slide was incubated with detection buffer.

### 2.7. Chick Chorioallantoic Membrane Assay 

The chick chorioallantoic membrane assay (CAM assay) was performed as described elsewhere [23]. Briefly, fertilized eggs (*n* = 10 per group) were incubated at 38 °C in an egg incubator for nine days. Eggs were candled to verify the viability of each embryo and to locate blood vessels suitable for injection. Growth factors were applied directly on the membrane in a relatively avascular region. Phosphate-buffered saline (PBS) and VEGF-A (100 nM) served as respective negative and positive controls. After the treatment, eggs were placed in the incubator for 48 h, and the angiogenic response was visually scored using a dissecting microscope. This study was performed double-blinded, and representative micrographs are shown.

### 2.8. Quantitative Real-Time PCR

Total RNA was isolated with TRIzol (Invitrogen, Waltham, MA, USA), and cDNA synthesis was performed using a high-capacity cDNA reverse transcription kit (Applied Biosystems, Waltham, MA, USA). Quantitative PCR was performed in triplicate with SYBR® Green PCR Master Mix (Applied Biosystems, Waltham, MA, USA) and was detected by CFX ConnectTM System (Bio-Rad, Hercules, CA, USA). PCR conditions were 30 cycles of 94 °C for 15 s, 52 °C for 30 s, and 72 °C for 30 s. Data were analyzed using CFX Manager Software (Bio-Rad, Hercules, CA, USA). Experimental Ct values were normalized to MRPL32. The sequences of PCR primers (5′-3′) were CGCGGTCGCCAAAAAGAAAG and TACTTGCAGTCGGCTCCAAAC for MDK; CCTTATCAAGATGCGAACTCACA and CGGAGGTGTTCTATGAGCTGG for HIF-1α; CGGAGGTGTTCTATGAGCTGG and AGCTTGTGTGTTCGCAGGAA for HIF-2α; CGGAGGTGTTCTATGAGCTGG and AGCTTGTGTGTTCGCAGGAA for MRPL32.

### 2.9. Transfection and Establishment of Stable Cell Lines

For transient overexpression or knock down, H1299 and A549 cells at 40% density were transfected with plasmids (2 μg per 60 mm dish) or siRNAS (80 nM), using Lipofectamine® 2000 (Life Technologies, Carlsbad, CA, USA). The transfected cells were allowed to stabilize for 48 h before the experiments.

### 2.10. Cell Viability and Colony-Forming Assays 

For cell viability assay, 2 × 10^3^ cells were plated in 96-well plates and incubated with the medium containing with drugs for 24/48 h. Cell viability was measured using CellTiter 96^®^ AQueous One Solution Reagent (Promega, Madison, WI, USA) containing tetrazolium compound [3-(4,5-dimethylthiazol-2-yl)-5-(3-carboxymethoxyphenyl)-2-(4-sulfophenyl)-2H-tetrazolium, inner salt; MTS] at 490 nm for 90 mins. Six replicate wells were used for each analysis, and at least three independent experiments were performed.

### 2.11. Immunohistochemistry Staining

Paraffin-embedded sections (5 μm thick) were deparaffinized, and heat-induced epitope retrieval was performed using Target Retrieval Solution 9 (Dako, Carpinteria, USA). The slides were treated with 3% hydrogen peroxide for 20 min for endogenous peroxidase activity, followed by washing in deionized water for 2–3 min. The slides were then incubated with 0.5% BSA blocking solution for 1 h at RT and then incubated with primary antibodies against CD-31, NF-κB, MDK, and T-antigen overnight in a 4 °C chamber. Immunoreactions were detected using the VECTASTAIN ABC HRO kit (Vector Laboratories, Burlingame, CA, USA). Hematoxylin was used as a counterstain. 

### 2.12. Immunoblotting

Proteins obtained from cell extracts were separated by SDS/polyacrylamide gel electrophoresis (PAGE) and then transferred to Immobilon-P membranes (Millipore, Bedford, MA, USA). Membranes were blocked with 5% w/v nonfat milk, incubated overnight at 4 °C with primary antibodies diluted 1:1000, and then incubated for 1 h with horseradish peroxidase-conjugated secondary antibodies. Antigen-antibody complexes were visualized using SuperSignal West Femto luminol enhancer solution (Thermo scientific, Waltham, MA, USA). Uncropped western blot can be found at Appendix A. 

### 2.13. Lung Orthotopic Mouse Model

All animal procedures were performed in accordance with a protocol approved the Institutional Animal Care and Use Committee (IACUC) of National Cancer Center Research Institute (NCCRI, Goyang-si, South Korea). NCCRI is an Association for Assessment and Accreditation of Laboratory Animal Care International (AAALAC International)-accredited facility and abides by the Institute of Laboratory Animal Resources (ILAR) guide and Usage Committee (NCC-14-255). Nude mice (BALB/cAnNCrj-nu/nu) from Charles River Laboratories Japan (CRLJ, Shin-Yokohama, Japan) were anesthetized with isoflurane via inhalation in an enclosed box chamber. Mice were positioned in a supine position, and the jaw and tongue were drawn away from the esophageal region using forceps while inserting a 22-gauge Hamilton TLC syringe (Model # 1705, Hamilton, Reno, NV, USA) into the trachea. Glass light was administered on the mouse’s upper chest and injected with 1 × 10^6^ cancer cells suspended in 50 μL of PBS. After instillation, the mouse was allowed to recover under visual control before placement back into the cage for a predetermined period after exposure.

### 2.14. Tumor Xenografts in Mice and the Spontaneous Lung Metastasis Model

Nude mice (BALB/cAnNCrj-nu/nu) from Charles River Laboratories Japan (CRLJ, Shin-Yokohama, Japan) were injected at a dorsal flank site with 1 × 10^6^ cancer cells suspended in 100 μL of PBS. Tumor volume was measured with calipers (volume = L × w × w × 0.52, where L is the width at the widest point of the tumor and w is the width perpendicular to L). When tumors reached a volume of 80–100 mm^3^ (termed day 0 in our experiments), tumor volume was measured once every three days. At the end of the experiment, mice were sacrificed by CO_2_ asphyxiation. Excised tumors were cut into two parts, and the tissues were fixed with 4% buffered formalin or frozen in liquid nitrogen.

For a spontaneous lung metastasis model, after performing the above-mentioned tumor xenograft model, solid tumors in mice flanks were surgically removed when the tumor volume reached 500~600 mm and were then treated with prepared drugs for one month. Lung metastasis was confirmed by luciferase imaging.

### 2.15. Statistical Analysis

Each result is expressed as the mean with standard error (SE) or standard deviation (SD) from >3 independent samples, as calculated with Microsoft Excel 2010 and SPSS statistical software packages. Groups were compared using two-tailed, unpaired Student’s *t*-test for all the assays, and statistical tests were two-sided. Nonparametric statistical tests used in each case were Mann–Whitney or chi-square tests. Differences were considered significant when *p* < 0.05 (*). 

## 3. Results

### 3.1. MDK Expression is Dependent on HIF-1α in NSCLC Cells

Hypoxia is a very important communicator between cancer cells and stromal cells. During reduced O_2_ tension, hypoxia acts as a stimulus for the activation of hypoxia-inducible factors, HIF-1α and HIF-2α [3]. Even though it is well known that almost all cancer cells express HIF-1α and HIF-2α under hypoxic conditions, HIF-2α is essential in colon cancer growth and progression among HIFs [24]. To screen which cancer cells were affected by HIF-1α or HIF-2α, we measured the protein and mRNA levels of HIF-1α and HIF-2α in H1299, A549 (NSCLC); Hep3B, HepG2 (HCC); HCT116, CT26 (Colon cancer); MCF-7, MDAMB231 (Breast cancer); MKN1, MKN5 (Gastric cancer); U87MG, SHSY5Y (Brain cancer); and SKOV3, SNU840 (Ovary cancer) under normoxic and hypoxic conditions (Appendix A) and confirmed hydroxyl-HIF-1a levels (Pro564 and Pro402), which is the essential approach of HIF-1α functional assessment (Appendix A). There were no cells with hypo- or hyper-expressed protein and mRNA levels of HIF-1α and HIF-2α. Since we could not identify which cancer cells were affected by HIF-1α or HIF-2α by measuring the expression levels of HIF-1α and HIF-2α, we determined that the cell viability decreased by knock down of HIFs under hypoxic conditions (Figure 1A); we confirmed HIF protein levels decreased by HIF-specific siRNA in Western blotting such as HIF-1α (Appendix A) and HIF-2α (Appendix A). Cell proliferation of H1299, A549 (NSCLC), MCF7, MDAMB231 (Breast cancer), MKN1, MKN45 (Gastric cancer), and U87MG, SHSY5Y (Brain cancer) cells was dependent on HIF-1α, and HCC cell proliferation was dependent on HIF-2α under hypoxia (Figure 1A,C). In addition, there were no significant cell viability differences between HIF-1α and HIF-2α in SKOV3, SNU840 (Ovary cancer), and CT26, HCT116 (Colon cancer) (Figure 1B). These results suggest that the NSCLC, breast, gastric, and brain cancer cells were dependent on HIF-1α levels and HCC cells were affected by HIF-2α. HIF-1α as the communicator between cancer cells and stromal cells exerts more influence than HIF-2α because the number of cancer cells dependent on HIF-1α was double that in cells dependent on HIF-2α in terms of viability. 

We chose NSCLC cells; H1299 was the most affected in terms of cell proliferation by HIF-1α knockdown. To examine which cytokine secretion from H1299 was dependent on HIF-1α, we collected CM from H1299 cells which were transfected with HIF-1α under normoxic conditions and from cells treated with siHIF-1α under hypoxic conditions and analyzed their respective cytokine profiles using a cytokine-profiling array. Figure 1D shows the top 14 cytokines that showed differences in their expression values between HIF-1α -expressed cells under normoxic conditions and HIF-1α knock down cells under hypoxic conditions. From this list of cytokines, we selected MDK (indicated *) for further analysis since it showed the highest difference between the two groups. To confirm the cytokine-profiling array results, we analyzed MDK mRNA and protein expression levels in H1299 and A549 cells which were transfected with HIF-1α and HIF-2α under normoxic conditions, as well as siHIF-1α and siHIF-2α expression under hypoxic conditions. Paul R et al. demonstrated that HIF-1α enhanced the transcription of MDK, acting on HIF-1α regulatory elements located in the MDK gene promoter [25]. MDK levels of A549 cells were very low or negative in the study conducted by Hao et al [21]. To confirm MDK levels, we performed Western blotting and QPCR with H1299, A549, H441, and H520 under normoxic and hypoxic conditions. The MDK levels of H1299 and A549 were much lower than those in H441 and H520 under the normoxic condition. However, the MDK levels in all cell lines showed similar levels under the hypoxic condition (Appendix A). HIF-1α expression under normoxic conditions increased MDK mRNA and protein levels, and HIF-1α -mediated the MDK level of mRNA and protein specifically decreased in response to siHIF-1α expression under hypoxic conditions (Figure 2A–C). These results suggest that MDK is regulated by HIF-1α, one of the down-stream proteins dependent on HIF-1α in NSCLC cells.

### 3.2. MDK Promotes the Migration of Endothelial Cells and Angiogenesis

To elucidate the physiological role of MDK in communication with stromal cells, we performed a migration assay with CCD-18Lu (fibroblast), THP-1 (macrophage), and HUVEC (endothelial cells). HUVEC alone showed migration response by CM derive from H1299 and A549 cells expressing MDK. We confirmed MDK overexpression by using western blotting (Appendix A). In contrast, all three stromal cells were migrated by CM from H1299 and A549 cells transfected with siCon under hypoxic conditions. Only HUVEC cells among the three stromal cells were abrogated by CM from H1299 and A549 cells transfected with siMDK (Figure 3A). The movement of HUVEC cells alone was significantly affected by MDK after crystal violet staining by measuring absorbance values at 540 nm (Figure 3B). To further substantiate previous results, we investigated the effects of MDK on VEGF-induced angiogenesis in vivo. We first focused on primary network formation in CAM assays. We used fertilized special pathogen-free eggs and challenged them, with CM from H1299 and A549 cells expressing pcDNA, pMDK, rhVEGF-A, and rhMDK. This experiment was performed in a double-blinded manner and after five days, the angiogenic response was visualized and scored (Appendix A and Figure 3C). rhMDK treatment promoted neovascularization, leading to the formation of a primary network, just like in the presence of the rhVEGF-A positive control, and CM from H1299 and A549 expressing MDK also significantly increased the neo-angiogenic response compared with CM from H1299 cells expressed pcDNA. In addition, even though hypoxia induced MDK and VEGF-A expression, MDK knock down by siMDK under hypoxic conditions did not decrease VEGF-A levels (Appendix A). These results indicate that MDK secreted from NSCLC cells under hypoxic condition promotes the migration of endothelial cells and neo-vascularization, and there was no correlation between MDK and VEGF-A.

### 3.3. MDK Overexpression Contributes to EMT in NSCLC Cells

MDK promotes EMT in pancreatic cancer [26]. Therefore, we performed migration assays with H1299 and A549 cells after transfection with siCon and siMDK under hypoxic conditions. Even though H1299 is a mesenchymal type and A549 is an epithelial type cell confirmed by short tandem repeat profiling, basal levels of E-cadherin were different between these cell lines (Figure 4 and Appendix A). H1299 and A549 cells under hypoxic conditions significantly migrated to the bottom chamber in transwell membranes compared with the control group, which was attenuated by siMDK (Figure 4A). These results suggest that the EMT character or cell migration of mesenchymal and epithelial type lung cancer cell lines is affected by hypoxia-induced MDK. To identify all the molecular alterations that MDK promoted during the migration of NSCLC cells, we analyzed by western blotting under the same condition as shown in Figure 4A. MDK expressed under hypoxia decreased Notch2 levels and induced Notch2 intracellular domain (NICD) cleavage, which is the active form of Notch2. The MDK-mediated increase in the Notch2 ICD level could increase NF-κB protein expression, which is known to be the central regulator of EMT and anti-apoptotic signaling, and as expected we found elevated NF-κB protein levels. To further validate MDK-mediated Notch2 cleavage and activation, we analyzed the protein expression of the well-known Notch target in mammals, Hes-1, and found elevated protein levels in NSCLC cells under hypoxic conditions. Furthermore, not only did we find substantially decreased E-cadherin and elevated Vimentin and Snail levels, but we also noticed elevated levels of Survivin and XIAP anti-apoptotic factors, increased phosphorylation of PI3K and AKT, and decreased levels of a pro-apoptotic factor, BAD, in hypoxia-exposed cells. However, under the same condition of H1299 and A549 cells, MDK knock down by siMDK upregulated the Notch 2 “level” that was not “signaling”. Actually, siMDK “downregulated” or “suppressed” Notch2 signaling. It was also noted that in MDK-induced NF-κB, Hes-1 survival signals, anti-apoptotic factors, and EMT markers were decreased. E-cadherin and BAD proteins were elevated in an inverse manner (Figure 4B–E). However, overexpressed survivin could not increase E-cadherin again, which was decreased by siMDK under hypoxic conditions (Appendix A). Hao et al. also reported iMDK, an MDK inhibitor, suppressed PI3K and induced the apoptotic pathway in H441 lung adenocarcinoma cells [21]. These results suggest that MDK increased by hypoxia induced the cleavage of Notch2 ICD, which in turn activated NF-κB and Hes-1. The MDK-Notch2-NF-κB-Hes-1 signal axis increased the expression of mesenchymal markers, and lung cancer cells became more disorganized and moveable. However, MDK knock down even under hypoxic conditions down regulated the Notch2-NF-κB-Hes-1 signal axis.

### 3.4. MDK Inhibitor Exerts Anti-Tumor and Anti-Metastasis Effects in an In Vivo Xenograft Model

To confirm the in vivo benefit of inhibiting MDK in an MDK-expressed lung tumor model, we constructed a xenograft in vivo model with H1299 cells, which stably expressed the empty vector (E.V.) and MDK genes because all tumor tissue could be exposed by hypoxia and we needed to check the MDK effects of the -pro-tumor, EMT, and angiogenesis via gain-of-function study (Figure 5A and Appendix A). MDK has been identified as a positive regulator of angiogenesis and is secreted by cancers to stimulate normal endothelial cell growth through paracrine signaling [27,28,29,30]. As mentioned before, the tumor volume of the stably overexpressed MDK group began to grow bigger than that of the control group treated with DMSO from around 40 days. Moreover, we applied the MDK inhibitor iMDK to determine the possibility of a therapeutic target. Hao et al. reported that iMDK suppressed endogenous MDK expression in lung adenocarcinoma cells and reduced tumor volume derived from H441 lung adenocarcinoma cells in a xenograft mouse model. iMDK (9 mg/kg) was intraperitoneally applied to one group for three days per week and the other group for five days per week, and the dosage ws determined after analysis of mice weight, serum AST, and ALT from mice treated with iMDK (9 mg/kg) [21]. In order to determine whether systemic administration of iMDK suppresses tumor growth, iMDK (9 mg/kg) was intraperitoneally injected once every two days into nude mice after a tumor volume around 100 mm^3^ was reached. Tumor volume and tumor weight in cells expressing MDK were almost twice as high as the control group. However, iMDK showed an anti-tumor effect which significantly decreased tumor volume and tumor weight compared to the control group (Figure 5–C). To substantiate the results with anti-tumor and anti-angiogenic effects, tissue staining assays were performed with H&E, anti-T antigen, Ki-67, and CD-31. Histological analysis showed a more proliferative, aggressive, and invasive phenotype in the MDK-expression group which had inter-mingled with the surrounding acinar tissue and more neo-vasculazation compared to the control group, whereas the majority of iMDK-treated tumors under the same condition were predominantly encapsulated or micro-invasive and the numbers of CD-31 stained cells were lower than in the control group. This aggravated invasive phenotype was readily revealed by immunohistochemical staining for the SV40T antigen oncoprotein, which is a marker of cancer cells (Figure 5C). In addition, we confirmed whether in vitro signaling occurred in vivo samples. Immunoblotting samples from the in vivo xenograft model showed that MDK-expression samples showed increased XIAP and vimentin and decreased E-cadherin cleaved caspase-3, which were reversed by iMDK treatment (Appendix A). To further determine the consequences of the invasiveness effect, we next investigated the possible occurrences of distant metastasis, mainly focusing on the lung. We first constructed an in vivo xenograft model, similar to Figure 5A, and then solid tumors in the mice flank were surgically removed. After 60 days, micro invasiveness of H1299 under MDK expression showed high metastasis in the lung compared with the control group. However, the iMDK treatment group had no significant metastasis compared with the MDK group. The presence of tumor metastasis in the lung was determined by lung images and corroborated by H&E staining (Figure 5D). Quantification of the incidence of macroscopic lung metastasis in each group revealed that the incidence of lung metastasis was 9-fold higher in the MDK-expression group than in the control group or iMDK-treated group. These results suggest that pro-tumor, EMT, and angiogenesis of lung cancer are dependent on MDK expression.

### 3.5. iMDK Suppresses MDK-Induced Lung Tumor Growth, Angiogenesis, and Metastasis

To further extend our previous in vivo results to study the suppressive effects on tumor growth, angiogenesis, and metastasis by iMDK treatment, H1299 cells expressing E.V. and MDK plasmid were inoculated by intratracheal injection (Figure 6A). We confirmed that our intratracheal injection did not result in leakage of the cells in the chest cavity through bio-luminescence images just after cell inoculation (Appendix A). The lung orthotopic model continued to grow in the control group (DMSO-treated group) and lung tumor colony numbers and lung weight in MDK-expressing H1299 cells were significantly higher than in the control, while lung tumor growth was obviously arrested in iMDK-treated groups. The lung colony number and weight in the iMDK-treated group were significantly lower than those in the MDK-expression group (Figure 6B–D). CD-31 and NF-κB were observed in higher numbers in the MDK-expression group than in the control group but not in the iMDK-treated group (Figure 6E). These results showed that MDK plays a critical role in lung tumorigenesis through induced angiogenesis and NF-κB activation, which was abrogated by iMDK treatment. Consistent with these results, the MDK-expression group in the lung orthotopic model H1299 cells that contained a stably transfected luciferase plasmid appeared markedly aggressive in luciferase images and had a significant survival disadvantage. On the other hand, the MDK+iMDK treatment group displayed greater effects of anti-tumor and -metastasis compared with the MDK-expression group and showed 60% survival rates until termination of the experiment at 100 days (Figure 7A,B). An increased tumor burden and metastatic secondary tumors in the MDK-expression group was also reflected by photon emission values of each mouse compared with the control and MDK+iMDK treatment. The MDK-expression group also showed a significantly higher pro-tumor effect and metastasis than the control group, which was abrogated by iMDK treatment (Figure 7C). We confirmed the primary site in the lung and the metastasis site in the liver through immunohistochemical staining with H&E and MDK (Appendix A). These in vivo results indicated that iMDK, an MDK target inhibitor, is a likely promising therapeutic anti-tumorigenic, -EMT, and -angiogenic drug target in lung cancers.

## 4. Discussion

HIFs play a critical role in the regulation of oxygen homeostasis. Within hypoxic cells, the diminished levels of oxygen stabilize HIF proteins and activate the HIF-dependent transcription, leading to the regulation of the expression of many genes [31]. The extensive transcriptional response regulated by HIFs involves the induction of genes for glucose metabolism, cell growth, apoptosis, angiogenesis, extracellular matrix remodeling, and metastasis [32]. HIF transcription factors form heterodimers composed of α (HIF-1α, 2α and 3α) and β (Arnt-1, 2 and 3) subunits [33]. Of these isoforms, HIF-1α and HIF-2α play pivotal roles in the cellular response to the lack of oxygen and tumor promotion. Despite similar properties, these isoforms might have some unique characteristics [31]. The experiments with targeted disruptions of both genes in mice resulted in embryonic lethality; however, the phenotypes in both cases were different [34]. HIFs are broadly expressed in human cancer cells and HIF1α and HIF2α were previously suspected of promoting tumor progression through overlapping functions [35]. All the cancer cells used in our study, H1299, A549 (NSCLC), Hep3B, HepG2 (HCC), HCT116, CT26 (Colon cancer), MCF-7, MDAMB231 (Breast cancer), MKN1, MKN5 (Gastric cancer), U87MG, SHSY5Y (Brain cancer), and SKOV3, SNU840 (Ovary cancer), expressed mRNA and proteins of HIF-1α and HIF-2α (Appendix A) under hypoxic conditions. However, their cell viability was different under conditions of HIF-1α or HIF-2α knock down. The cell viabilities of NSCLC, breast, gastric, and brain cancer cells were decreased more by HIF-1α than in the case of HIF-2α knock down. Conversely, HCC cells were affected more by HIF-2α than by HIF-1α. There were no large differences in cell viability between HIF-1α and HIF-2α in ovary and colon cancer cells (Figure 1A–C). In particular, depletion of HIF-1α had the greatest effect on the proliferation of H1299 cells in HIF-1α-dependent cells compared with depletion of HIF-2α. These results suggested that HIF-1α and HIF-2α are universally expressed in a variety of cancers. HIF-1α plays a dominant role in tumor progression and appears to be an obvious significant prognostic factor in NSCLC [36].

Hypoxia or HIF expression plays pivotal roles in tumor microenvironment (TME) communication. The H1299 cell in NSCLC cells was the most affected in terms of cell viability by HIF-1α knock-down. To extend our knowledge about the secretome between HIF-1α expression under normoxic conditions and HIF-1α knock down under hypoxic conditions in H1299 cells used in this study, we profiled the top 14 cytokines secreted from HIF-1α-induced H1299 cells and selected MDK, which showed the greatest differences between two samples, for further investigation by examining their effects on the tumor microenvironment (Figure 1D). The expression and secretion of MDK were clearly dependent on HIF-1α in NSCLC cells (Figure 2). MDK is a secreted heparin-binding growth factor identified as a product of a retinoic acid response gene [37,38]. The patho-physiological effects of MDK include enhanced anti-apoptotic activity [39] and angiogenic activity [30]. Our results showed that MDK-expressing H1299 and A549 cells increased survivin and XIAP levels, anti-apoptotic molecules, and decreased BAD levels, which are pro-apoptotic proteins, consistent with previously published NSCLC data on MDK in the study conducted by Hao et al. [21] (Figure 4E). In the angiogenesis study, CM from hypoxia-induced MDK in H1299 and A549 cells showed elevated endothelial cell migration, but not in fibroblasts and macrophages, and increased CAM vasculogenesis. In contrast, MDK knock down only decreased HUVEC cell movement even under hypoxic conditions (Figure 3). Stable MDK-expressing H1299 cells promoted higher lung tumor colony numbers and lung weight by increased CD-31 and NF-κB compared to those in control cells in the lung orthotopic mice model. In addition, hypoxia-induced MDK affects cancer cells through an autocrine mechanism. In our study, MDK-expressing H1299 and A549 cells increased the Notch2-NF-κB-Hes1 signaling axis. MDK-induced NF-κB also increased cancer cell proliferation through PI3K-Akt signaling and Hes1 triggered epithelial-to-mesenchymal transition via increased Vimentin and Snail levels, and decreased E-cadherin levels, which were not recovered by survivin expression under MDK knock down and hypoxia (Appendix A). Consequently, the migration of MDK-induced H1299 and A549 cells was significantly increased more than that in the control group through the Notch2-NF-κB-Hes1 axis (Figure 4). These results suggest that MDK induced by hypoxia in NSCLC cells plays two major roles such as tumor progression and EMT in cancer cells, and MDK increased endothelial cell migration and angiogenesis in the tumor microenvironment. 

To demonstrate that MDK could be a therapeutic target, we studied MDK gain and loss of function in H1299 and A549 cells. The cancer cell movement induced by MDK was significantly abrogated by siMDK since MDK-Notch2-NF-κB-Hes1 signaling was diminished by MDK knock down (Figure 4). In an in vivo lung orthotopic model, MDK-induced tumor progression decreased due to a decrease in CD-31 and NF-κB expression in response to iMDK treatment. Similarly, in studies involving an in vivo xenograft model, iMDK significantly decreased tumor volume and tumor weight. Notably, the MDK-expression group promoted tumor volume and also micro-invasiveness of cancer cells measured via H&E and anti-T antigen staining. However, iMDK exerted an anti-angiogenetic effect and most of the cancer cells were encapsulated (Figure 5). In an in vivo spontaneous lung metastasis model and lung orthotopic model, the MDK-expression group showed 90% lung metastasis rates and metastatic luciferase images and photon emission, but not in the case of iMDK treatment (Figure 5D,E, Figure 7B,C). These results suggested that the strategy of MDK inhibition might be a promising therapy for NSCLC cells in terms of anti-angiogenic therapy.

## 5. Conclusions

HIF-1α has a dominant role in NSCLC via MDK expression, which promotes cancer cell progression and EMT through the Nothc2-NF-κB-Hes1 signaling axis through autocrine signaling and the migration of endothelial cells and vascuolization via a paracrine mechanism. However, MDK knock down suppresses the pro-tumor effect of hypoxia-induced MDK in vitro. Moreover, the MDK inhibitor decreased tumor progression, metastasis, and angiogenesis in vivo. Targeting the expression of MDK provides a new therapeutic approach for the treatment of MDK-expressing NSCLC patients and is useful in other types of cancers that express MDK (Figure 7D).

## Figures and Tables

**Figure 1 cancers-12-02402-f001:**
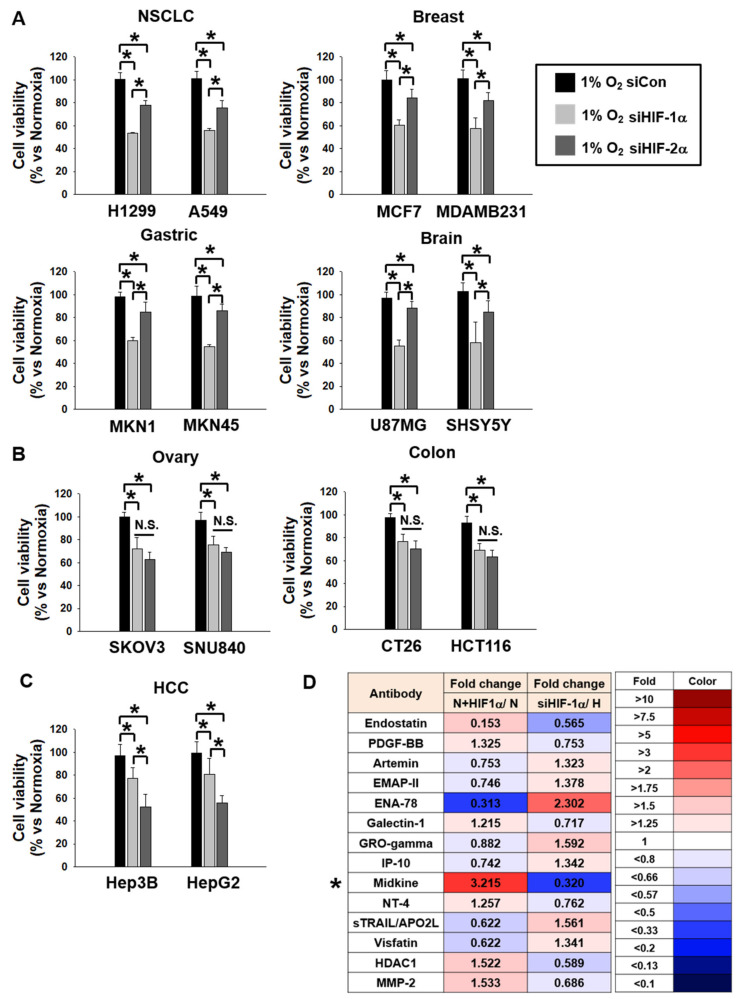
Cell lines were more dependent on HIF-1α or HIF-2α. (**A**–**C**) HIF-1α and HIF-2α were knocked down, and then cell viability was measured. H1299, A549 (NSCLC); Hep3B, HepG2 (HCC); HCT116, CT26 (Colon cancer); SKOV3, SNU840 (Ovary cancer); MCF-7, MDAMB231 (Breast cancer); MKN1, MKN45 (Gastric cancer); and U87MG, SHSY5Y (Brain cancer) cells were incubated for 16 h under normoxic (20% O_2_) or hypoxic (1% O_2_) conditions after transfection with siCon, siHIF-1α, and siHIF-2α. Cell viability was measured using MTS at 56 h after normoxic and hypoxic conditions were imposed. Student’s *t*-test, average ±SD; *n* = 6, * *p *< 0.05. (**D**) List of the top 13 cytokines with increased or decreased levels of secretion following pcDNA, HIF-1α, siCon, and siHIF-1α transfection with H1299 cells in normoxic and hypoxic chambers.

**Figure 2 cancers-12-02402-f002:**
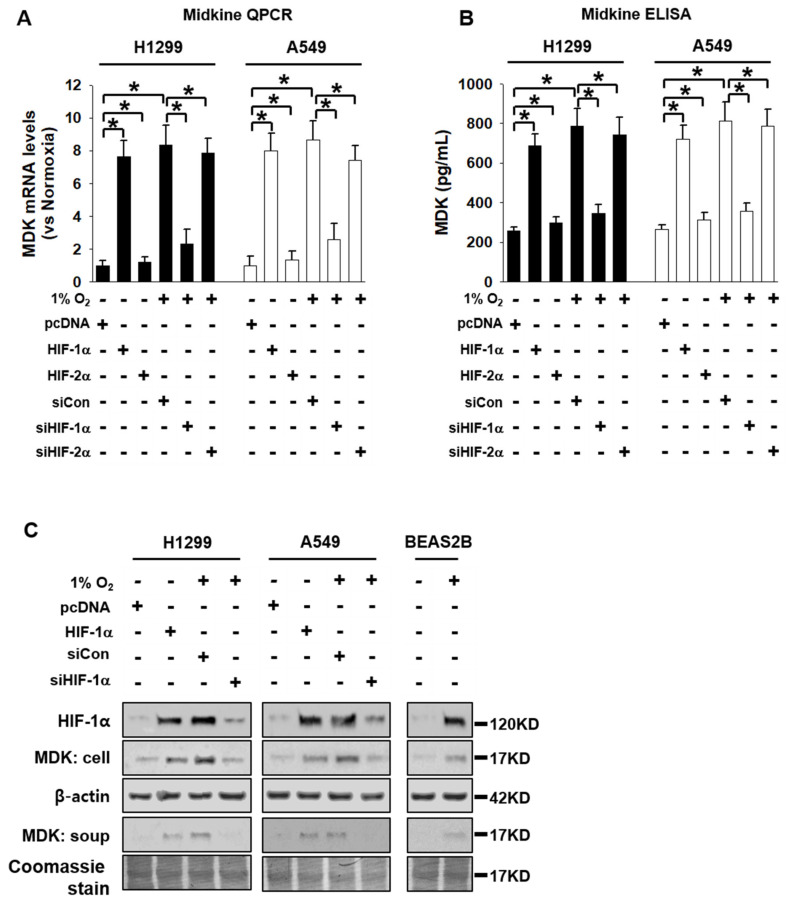
MDK levels were increased by HIF-1α in NSCLC cells. (**A**) MDK mRNA levels were increased by HIF-1α. H1299 and A549 cells were transfected with pcDNA, HIF-1α, HIF-2α, siCon, siHIF-1α, and siHIF-2α under normoxic and hypoxic conditions; then, MDK mRNA levels were measured by quantitative RT-qPCR. (**B**) and (**C**) Under the same conditions as in Figure 2A, MDK protein levels were analyzed using MDK ELISA assay and immunoblotting. Student’s *t*-test was performed and values displayed as average ±SD; *n* = 6, * *p *< 0.05. These cells and CM were harvested with lysis buffer and western blotting was performed.

**Figure 3 cancers-12-02402-f003:**
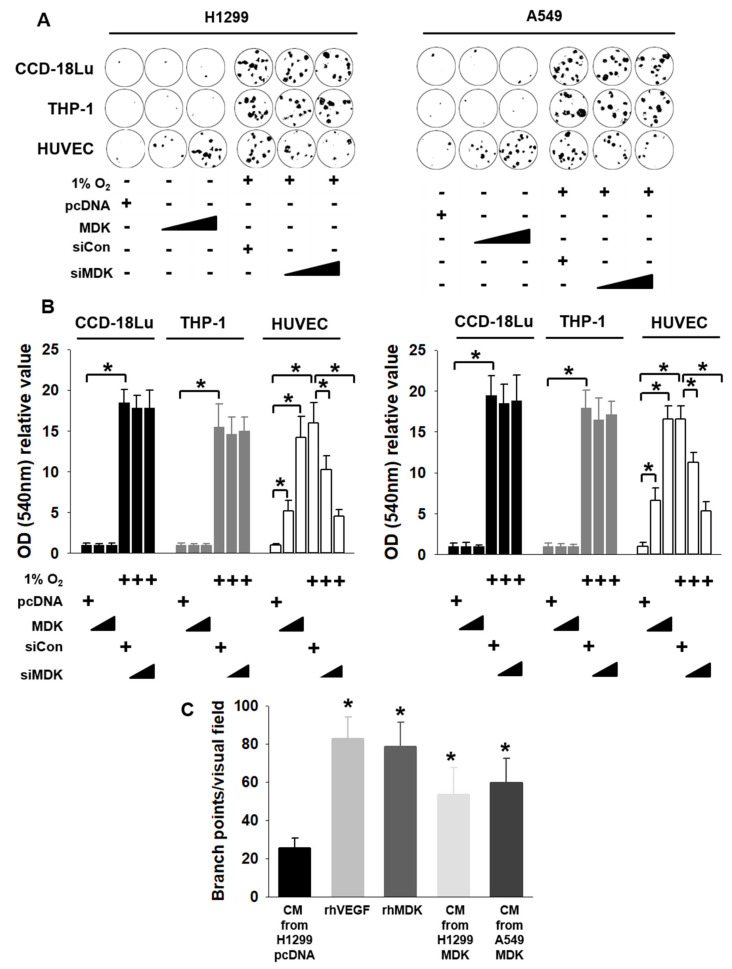
MDK promoted endothelial cell migration and angiogenesis in NSCLC cells. (**A**) Endothelial cells migrated in response to MDK overexpression. H1299 and A549 cells were transfected with pcDNA, MDK, siCon, and siMDK under normoxic and hypoxic conditions and then the media were collected. CCD-18Lu, THP-1, and HUVEC 5 × 10^4^ cells were plated in serum-free medium onto the upper chamber, and CM were contained in the bottom chamber to serve as a chemo-attractant; the cells were incubated at 37 °C in a CO_2_ incubator for 16 h. Transwell membranes were stained with crystal violet (CV) and viewed under a light microscope (magnification, ×100). (**B**) Relative numbers of cells that transmigrated through the membrane in Figure 3A were determined by staining cells on the undersurface of the transwell membrane with CV staining by cell lysis and measurement of absorbance values at 540 nm. Staining levels were directly proportional to the number of cells. Student’s *t*-test was used for statistical significance and the values plotted as the average ±SD; *n* = 6, * *p *< 0.05. (**C**) Angiogenic activity of MDK was measured using the chicken chorioallantoic membrane assay (CAM). CM were collected from H1299 and A549 overexpressing MDK. To check the angiogenic activity of MDK, PBS, rhVEGF165 (100 ng/mL, positive control), and rhMDK (100 ng/mL) were added to the CM from H1299 and A549 on day 0. The data are presented as the mean number of blood vessel branches Macroscopic ex ovo features of the CAM on day 5 of the incubation. Student’s *t*-test was performed for statistical analysis and data plotted as average ±SD; *n* = 4, * *p* < 0.05.

**Figure 4 cancers-12-02402-f004:**
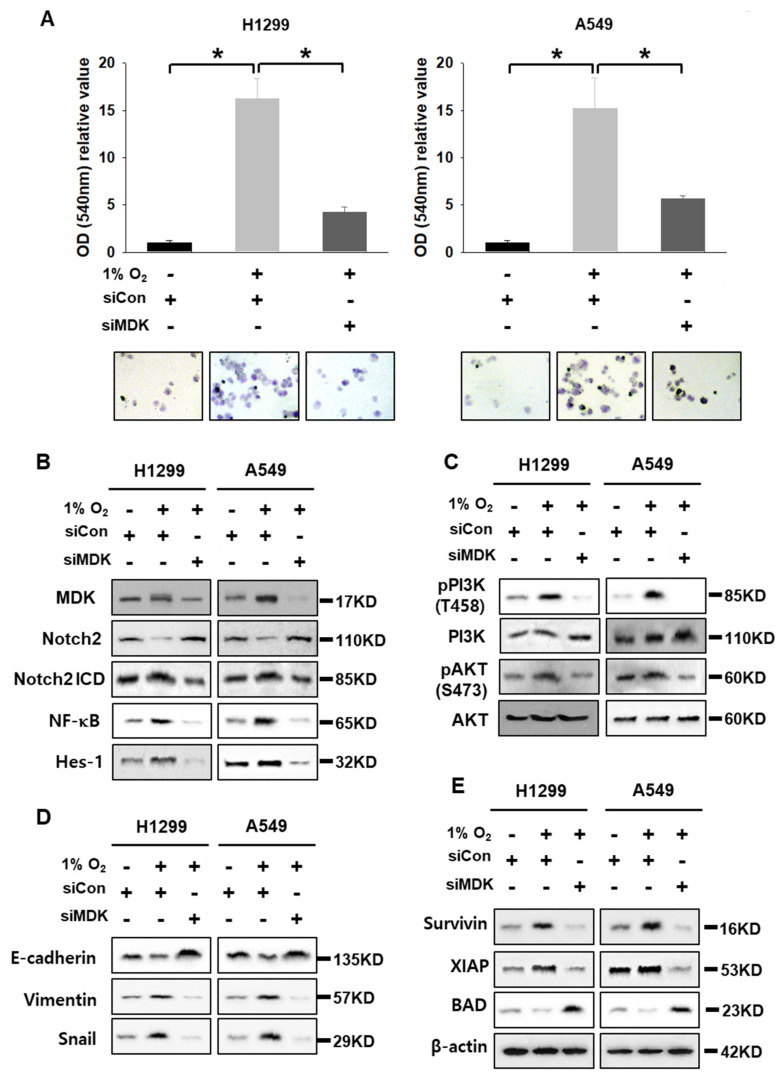
MDK knock down inhibits MDK-related signaling. siCon and siMDK were transfected with H1299 and A549 cells, and then cell migration and signaling were measured by immunoblotting assays after exposure to normoxic or hypoxic conditions. (**A**) H1299 and A549 cells were plated in serum-free medium onto the upper chamber, and 10% FBS medium was contained in the bottom chamber to act as a chemoattractant; the cells were incubated at 37 °C in a CO_2_ incubator for 16 h. Student’s *t* test was performed and the values displayed as average ±SD; *n* = 6, * *p *< 0.05. (**B**) Identification of MDK-Notch2-Hes-1 signaling. (**C**) Checking the PI3K-AKT signaling. (**D**) Examining the EMT-related proteins. (**E**) Checking anti-apoptotic and pro-apoptotic proteins.

**Figure 5 cancers-12-02402-f005:**
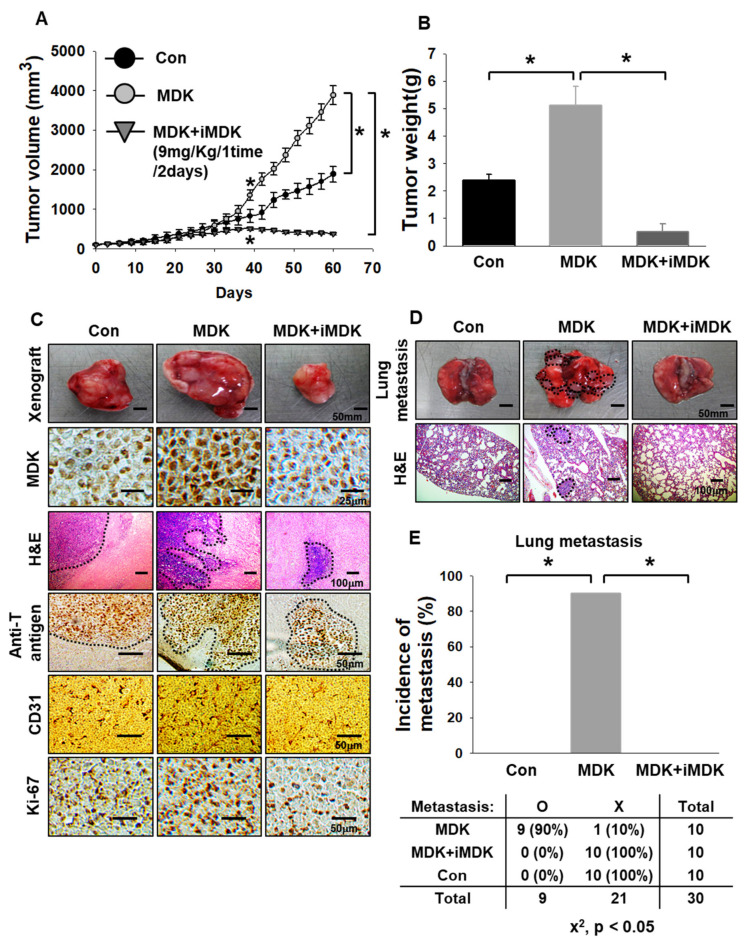
MDK inhibitor exhibits anti-tumor and anti-metastasis effects on a xenograft mouse model of NSCLC. (**A**) H1299 cells were transfected with E.V. and Myc-DDK-MDK, which were selected by G418 (400 μg/mL) until colony formation. H1299 cells were injected subcutaneously into nude mice 1 × 10^6^ cells/mouse, and tumor formation was measured every 3 days for 60 days from day 7 after implantation. Groups were intraperitoneally treated: DMSO (100 μL/ 1 time/ 2 days) in the control and iMDK (9 mg/ kg/ 1 time/ 2 days) in the drug treated-groups. Student’s *t* test was performed and values displayed as average ±SEM; *n* = 10, * *p *< 0.05. (**B**) Tumor weight was measured after removal. Student’s *t* test was performed and values displayed as average ±SEM; *n* = 10, * *p *< 0.05. (**C**) Tumor xenograft images, hematoxylin and eosin (H&E), and immunohistochemical staining of anti-MDK, anti-T antigen, and CD-31 in mice tumors from each group. (Scale bar: 25/ 50/ 100 μm). (**D**) Lung metastasis observed in lung tumor images and histological H&E staining of tissue sections from DMSO and drug-treated group (Scale bar: 100 μm). (**E**) Top: quantification of the incidence of mice with lung metastasis in the control, MDK, and MDK+iMDK-treated group. Bottom: contingency table relating the number and percentage of mice in each treatment or metastasis case. iMDK treated-mice showed a statistically significant decrease in the incidence of lung metastasis based on the chi-square test (*n* = 10, * *p *< 0.05).

**Figure 6 cancers-12-02402-f006:**
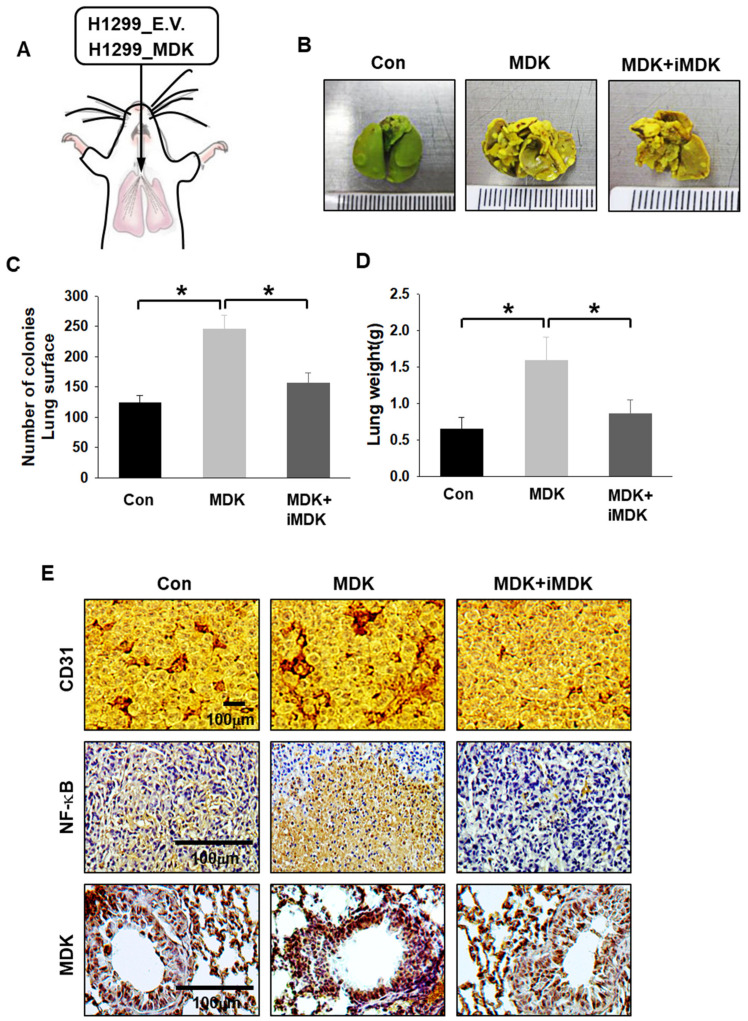
iMDK decreases MDK-induced lung tumorigenesis. (**A**) Schematic diagram of the experimental system to induce the lung orthotopic model in mice by intratracheal injection with the H1299 stable cell line. (**B**) Transfected cells were intratracheally injected into nude mice (1 × 10^6^ cells/mouse), and lung tumor images were measured two months after injection. The control and MDK expression groups were intraperitoneally treated with DMSO (100 μL/ 1 time/ 2 days) and iMDK (9 mg/ kg/ 1 time/ 2 days). At the first sign of morbidity, mice were euthanized and mice lungs were isolated and stained with Bouin’s fixative. (**C**) The number of colonies formed on the lungs in iMDK-treated mice was significantly lower than that in the MDK-expression group. Student’s *t*-test was performed and values displayed as average ±SEM; *n* = 5, * *p *< 0.05. (**D**). The tumor weight in iMDK-treated lungs was also smaller than in lungs of MDK-expressed H1299. Student’s *t* test was performed for statistical analysis and the values displayed as average ±SEM; *n* = 5, * *p *< 0.05. (**E**) Immunohistochemical staining of CD-31, NF-κB, and MDK in mice lung tumors from each group obtained at necropsy. (Scale bar: 100 μm).

**Figure 7 cancers-12-02402-f007:**
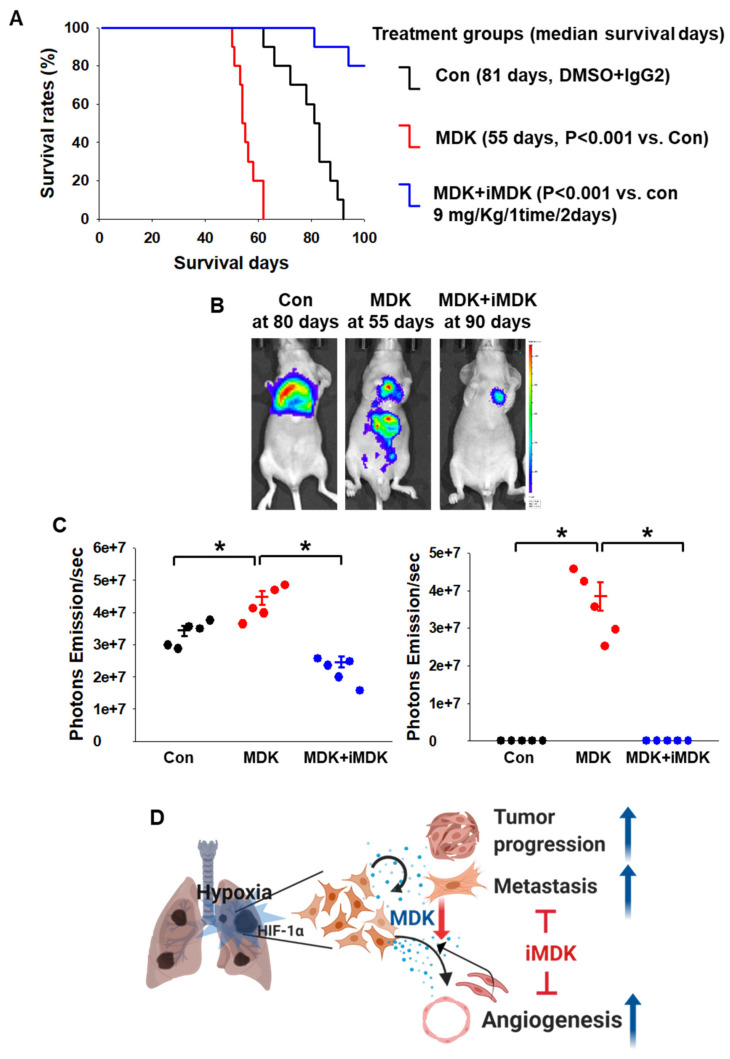
Increased life span and tumor reduction in iMDK-treated lung tumorigenesis. (**A**) Kaplan–Meier survival curves in lung orthotopic tumor-bearing mice treated continuously with control, MDK, and MDK+iMDK starting at 30 days after treatment. DMSO treated mice showed a median survival days of 81 days; mice receiving continuous MDK+iMDK treatment demonstrated a survival benefit but not in the MDK-expression group. Kaplan–Meier statistic, *n* = 5, * *p *< 0.001. (**B**) Representative bio-luminescence images at 80 days for the control, at 55 days for MDK, and at 90 days for iMDK are shown, revealing a BLI signal originating from the site of injection. (**C**) The photon emission values in Figure 7B represent the mean ±SEM of the indicated number of mice. * *p* < 0.05 based on Student’s *t*-test. (**D**) Schematic diagram summarizing hypoxia-induced MDK NSCLC cell tumor progression and metastasis and its inhibition.

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
