# Peer review of "Midkine Is a Potential Therapeutic Target of Tumorigenesis, Angiogenesis, and Metastasis in Non-Small Cell Lung Cancer"

_cancers, 2020, doi:10.3390/cancers12092402_

Round 1
Reviewer 1 Report
This study demonstrated that midkine (MDK) can be secreted from non-small cell lung cancer (NSCLC) cells in response to hypoxia (HIF1alpha). It was further shown that MDK enhances metastasis, angiogenesis, and tumor growth via signaling pathway associated with Notch2 and HES1. Authors concluded that MDK can be a potential therapeutic target of NSCLC.
Generally, experiments were performed well. However, effect of MDK on growth of NSCLC cells and tumor, including hypoxia-driven MDK expression have already been published (Hao et al. PLOS ONE, 2013; Zhang et al. Am J Phys Cell Physiol, 2014). Thus, in my view, novelty/originality of the present study is quite low. Additionally, above highly related published studies have not been cited in this manuscript.
Author Response
Review 1
Comments and Suggestions for Authors
This study demonstrated that midkine (MDK) can be secreted from non-small cell lung cancer (NSCLC) cells in response to hypoxia (HIF1alpha). It was further shown that MDK enhances metastasis, angiogenesis, and tumor growth via signaling pathway associated with Notch2 and HES1. Authors concluded that MDK can be a potential therapeutic target of NSCLC.
Generally, experiments were performed well. However, effect of MDK on growth of NSCLC cells and tumor, including hypoxia-driven MDK expression have already been published (Hao et al. PLOS ONE, 2013; Zhang et al. Am J Phys Cell Physiol, 2014). Thus, in my view, novelty/originality of the present study is quite low. Additionally, above highly related published studies have not been cited in this manuscript.
Thank you for your comments, I fully understand your concerns that novelty or originality of my manuscript is low. However, if we could review thoroughly my manuscript, we will find that this manuscript gets further study from Hao et al. and Zhang et al. First of all, the biggest finding is that Hao et al. screened MDK inhibitor and found iMDK which reduced endogenous expression of MDK. They performed anti-cancer study of iMDK with MDK-positive H441 and H520 lung adenocarcinoma cells in vitro and tumor growth in a xenograft mouse model in vivo. They performed in vivo experiment with two groups of control and iMDK. Although iMDK decreased lung cancer volume, they did not have MDK expressed tumor growth data. However, we performed in vivo experiment with three groups of control, MDK expressed, and MDK expressed+iMDK. The tumor volume and weight of MDK expressed group was much bigger than control and the volume and weight of MDK+iMDK group was much lower than even in control. We suggest that MDK is very important of lung cancer progression and also iMDK might be a potential therapeutic target of MDK expressed lung cancer in vivo model. In addition, we found the margin of MDK expressed tissue in xenograft model was very irregular and increased angiogenesis compared with control. To confirm whether MDK increases lung cancer metastasis or not, we performed tumor xenografts in mice and spontaneous lung metastasis model and lung orthotopic model. We proved MDK increased lung cancer metastasis and iMDK decreased metastasis compared with even in control group. Of course, we accepted that Zhang et al. approved MDK changed epithelial cells to cells of mesenchymal phenotype in vitro. The different thing is that we approved MDK increased angiogenesis through promoting endothelial migration, increased migration via Notch-NF-kB-Hes1 signaling in vitro, and increased metastasis in two in vivo models. Second, your another concern was that we did not cite these two highly related studies, Hao et al. and Zhang et al. Therefore, we did reference these two studies in introduction, results, and discussion.

Reviewer 2 Report
In my opinion the manuscript is well written and weel desgoned.
I have only some concerns about results and figures preseted.
For example image quality of transwell migration assay is very low so are not essentials.
On the contrary, the authors should show chicken chorioallantoic membrane assay images to better demonstrate their results.
Moreover immunoistochemical staining figures are not clear enough. Moreover, sections showed (CD31 and MDK) should be progressive with the same magnification to better highlight the result.
Author Response
Review 2
Comments and Suggestions for Authors
In my opinion the manuscript is well written and weel desgoned.
I have only some concerns about results and figures preseted.
For example image quality of transwell migration assay is very low so are not essentials.
On the contrary, the authors should show chicken chorioallantoic membrane assay images to better demonstrate their results.
Moreover immunoistochemical staining figures are not clear enough. Moreover, sections showed (CD31 and MDK) should be progressive with the same magnification to better highlight the result.
Thank you for your comments, we changed the images of transwell migration assay in Figure 4A. In addition, we also changed or replaced with clear and high resolution immunohistochemical staining images in Figure 5 and 6 as you mentioned.

Reviewer 3 Report
In this study, the authors identified that Midkine (MDK) regulates tumor progression and cancer metastasis in non-small cell lung cancer (NSCLC). The authors described that HIF-1α that have been identified and evaluate their transcriptions values in NSCLC than HIF-2α. They further found that MDK participates in the HIF-1α signature. The results also showed MDK can promote cell migration, angiogenesis, epithelial-mesenchymal transition and prevents apoptosis in vitro. Furthermore, several evidences showed that MDK inhibitors may be a promising therapeutic target in xenograft and lung orthotopic models. Although the findings are interesting, there are still some major flaws in the manuscript that need to be addressed carefully and thoroughly.
Major:
1. HIF-1α is a well-known transcription factor during tumorigenesis and can regulate various signaling molecules. The authors should explain why they chose the cytokines secretion part to investigate in this study. Moreover, the authors should describe in great details how HIF-1α regulates MDK mRNA and protein expression levels. Is there a hypoxia response element domain in the MDK promoter region that can be bound and transcriptionally activated by HIF1a binding?
2. In Figure 1, the authors examined the cell viability in various cancer cell lines, including lung, breast, gastric, brain, ovary, colon and HCC. The results performed showed that NSCLC, breast, gastric and brain cancer cells had similar results under normoxia or hypoxia conditions. However, the authors did not discuss why they then decided to use NSCLC for the remainder of the study. In addition, the authors should describe the cell lines on the heat-map in Figure 1D.
3. In several cell line models, the authors utilized HIF-1α or HIF-2α overexpression plasmids under normoxic conditions. This reviewer is worried that even HIF-1α/HIF-2α can be ectopically overexpressed, however, more proteins will still be degraded under normoxia status.
4. H1299 has been confirmed as a malignant large cell lung cancer cell line that belongs to the mesenchymal-type (M). The author's study should use endogenous low or null E-cadherin protein expressed H1299 cells to do further experiments. The authors should discuss or provide cell line authentication certificate to verify their H1299 is indeed authenticate H1299 cell line.
5. Similarly, H1299 (M type) and A549 (E type) are mesenchymal type and epithelial type, respectively. However, the authors performed the experiments and showed the same trends in figure 4A. This phenomenon needs to be reanalyzed or discussed carefully for the plausible reasons.
6. What is the explainable mechanism of iMDK effects in NSCLC? The authors should validate the cytotoxicity of iMDK, cite previous studies and describe how to determine the appropriate dosage in animal experiments. A careful discussion should be addressed.
7. The authors mentioned they process the lung orthotopic model in mice by injecting lung cancer cell line. However, they should show the exisiting bio-luminescence images at day 0 to show the reader that indeed live cancer cells were injected orthotopically and no leakage of the cells in the chest cavity occurred. The authors should show the histopathological H&E staining images and immunohistochemical staining results including primary sites and metastasis sites with pictures taken at low power field and at high power field.
8. The authors purposed that Notch2, NF-kB, Hes-1 by MDK inductions in their model. Moreover, the authors also observed that PI3K/Akt, Survivin/XIAP/BAD pathways have been activated. However, the authors need to claim which is the main mechanism MDK utilizes to regulate these pathways in addition to also need to confirm which pathway plays the major role in EMT changes.
Minor:
1. The authors need to detect the hydroxylation status of HIF-1α in all protein experiments. Comparing HIF-1α and hydroxyl- HIF-1α levels is the essential approach of HIF1a functional assessment.
2. Poor organ appearance conditions and immunohistachemical staining results must be replaced with clear and high resolution photographs.
3. The authors should rearrange the IVIS images at the same magnification, and expand the scale of their bio-luminescence images.
Author Response
Review 3
Comments and Suggestions for Authors
In this study, the authors identified that Midkine (MDK) regulates tumor progression and cancer metastasis in non-small cell lung cancer (NSCLC). The authors described that HIF-1α that have been identified and evaluate their transcriptions values in NSCLC than HIF-2α. They further found that MDK participates in the HIF-1α signature. The results also showed MDK can promote cell migration, angiogenesis, epithelial-mesenchymal transition and prevents apoptosis in vitro. Furthermore, several evidences showed that MDK inhibitors may be a promising therapeutic target in xenograft and lung orthotopic models. Although the findings are interesting, there are still some major flaws in the manuscript that need to be addressed carefully and thoroughly.
Major:
- HIF-1α is a well-known transcription factor during tumorigenesis and can regulate various signaling molecules. The authors should explain why they chose the cytokines secretion part to investigate in this study. Moreover, the authors should describe in great details how HIF-1α regulates MDK mRNA and protein expression levels. Is there a hypoxia response element domain in the MDK promoter region that can be bound and transcriptionally activated by HIF1a binding?
Thank you for your comments, we mentioned that HIF-1a enhanced the transcription of MDK, acting on HIF-1a regulatory elements located in the MDK gene promoter from line 264 to 266 (Paul et al. JBC 2004).
- In Figure 1, the authors examined the cell viability in various cancer cell lines, including lung, breast, gastric, brain, ovary, colon and HCC. The results performed showed that NSCLC, breast, gastric and brain cancer cells had similar results under normoxia or hypoxia conditions. However, the authors did not discuss why they then decided to use NSCLC for the remainder of the study. In addition, the authors should describe the cell lines on the heat-map in Figure 1D.
Thank you for your detailed revision, we mentioned why we did choose NSCLCs, especially H1299 for cytokine array from line 264 to 269. In addition, we describe H1299 cell line used in Figure 1D from line 254 to line 255.
- In several cell line models, the authors utilized HIF-1α or HIF-2α overexpression plasmids under normoxic conditions. This reviewer is worried that even HIF-1α/HIF-2α can be ectopically overexpressed, however, more proteins will still be degraded under normoxia status.
Thank you for your comments. Although I understand your concerns that HIFs proteins were degraded under normoxia status, ectopically expressed HIFs proteins were still working over 24 h after transfection. Tan et al. Oncotarget, 2017 also did experiment that they measured BCL-9 (B-cell CLL/lymphoma 9 protein) expression of HIF-1a and HIF-2a downstream at 24 h after ectopically expressed HIF-1a and HIF-2a with HCT116 and SW-480.
- H1299 has been confirmed as a malignant large cell lung cancer cell line that belongs to the mesenchymal-type (M). The author's study should use endogenous low or null E-cadherin protein expressed H1299 cells to do further experiments. The authors should discuss or provide cell line authentication certificate to verify their H1299 is indeed authenticate H1299 cell line.
Thank you for your comment, H1299 cells have lower level of E-cadherin than that in A549 cell in Figure 4D as you mentioned and also checked cell line authentication of H1299 and A549 through short tandem repeat genotyping. We discussed this issue from line 336 to line 339.
- Similarly, H1299 (M type) and A549 (E type) are mesenchymal type and epithelial type, respectively. However, the authors performed the experiments and showed the same trends in figure 4A. This phenomenon needs to be reanalyzed or discussed carefully for the plausible reasons.
Thank you for your comments, H1299 and A549 are totally different character type cell. However, hypoxia-induced MDK increased cell migration in both cell lines, which was decreased by siMDK. These results suggest that the EMT character or cell migration of both type lung cancer cell lines are affected on hypoxia-induced MDK. We mentioned this from line 341 to line 342.
- What is the explainable mechanism of iMDK effects in NSCLC? The authors should validate the cytotoxicity of iMDK, cite previous studies and describe how to determine the appropriate dosage in animal experiments. A careful discussion should be addressed.
Thank you for your comments, Hao et al. (PLOSone, 2013) did screen 44,000 compounds to inhibit MDK expression. They found iMDK compound and did experiment iMDK inhibition effect in vitro and in vivo. Therefore, we did reference Hao et al. paper and describe how to determine the appropriate dosage in animal experiments from line 384 to line 389.
- The authors mentioned they process the lung orthotopic model in mice by injecting lung cancer cell line. However, they should show the exisiting bio-luminescence images at day 0 to show the reader that indeed live cancer cells were injected orthotopically and no leakage of the cells in the chest cavity occurred.
Thank you for your comments, we confirmed bio-luminescence images just after cell inoculation and mentioned this results from line 437 to line 439.
The authors should show the histopathological H&E staining images and immunohistochemical staining results including primary sites and metastasis sites with pictures taken at low power field and at high power field.
Thank you for your comments, we performed H&E staining and IHC staining of MDK with lung from primary site and liver from metastasis site in lung orthotopic model. We described these results from line 456 to line 457.
- The authors purposed that Notch2, NF-kB, Hes-1 by MDK inductions in their model. Moreover, the authors also observed that PI3K/Akt, Survivin/XIAP/BAD pathways have been activated. However, the authors need to claim which is the main mechanism MDK utilizes to regulate these pathways in addition to also need to confirm which pathway plays the major role in EMT changes.
Thank you for your comment, we identified that MDK increased hypoxia induced the cleavage of Notch2 ICD, which in turn activated NF-kB and Hes-1. These MDK-Notch2-NF-kB-Hes-1 axis increased the expression of mesenchymal markers. However, survivin expression under MDK knock down and hypoxia could not recover E-cadherin. Therefore, we mentioned that main signaling of MDK-induced EMT is Notch2-NF-kB-Hes1 axis from line 533 to line 534.
Minor:
- The authors need to detect the hydroxylation status of HIF-1α in all protein experiments. Comparing HIF-1α and hydroxyl- HIF-1α levels is the essential approach of HIF1a functional assessment.
Thank you for your comment, we performed western blotting of HIF-la hydroxyl Pro564 and 402 with all cancer cell lines in Figures S1 and mentioned these results from line 237 to line 238.
- Poor organ appearance conditions and immunohistachemical staining results must be replaced with clear and high resolution photographs.
Thank you for your comment, we changed or replaced with clear and high resolution immunohistochemical staining images in Figure 5 and 6 as you mentioned.
- The authors should rearrange the IVIS images at the same magnification, and expand the scale of their bio-luminescence images.
Thank you for your comments, we did regulate the same scale of bio-luminescence images in Figure 7 as you mentioned.

Reviewer 4 Report
Overall, the original research paper entitled “Midkine is a potential therapeutic target of tumorigenesis, angiogenesis and metastasis in NSCLC” is of some interest to the readers of Cancers, even though does not add much novelty compared to previously published data. It has several deficiencies, and requires major revisions.
A. As a major caveat, the major novelty is regulation of MDK by HIF1, as well as NF-kB/ Notch/EMT signaling data. The role of MDK in NSCLC, development of iMDK inhibitor to MDK (including xenograft studies), and even signaling / apoptosis regulated by MDK in NSCLC were all previously published by the Japanese group (Hao et al, Naomoto - PLoS One 2013 – which is actually cited by you in REF 30- with an incorrect format (first names instead of last names in that REF) and in another paper by the same group, Masui M et al Anticancer Res 2016).
B.Abstract:
“Cells were incubated under hypoxic conditions”: change to “Cancer cells were incubated…”
“…the viability ..showed more sensitivity to HIF-01 that HIF-2”- could you rephrase/ perhaps say that “cell proliferation of XXX cells (? Under hypoxia) was more dependent on HIF1, except HCC cells where it was more dependent on HIF2”?
I would add a few words on why NSCLC were chosen for further study and not other cell types (more suppression of proliferation after HIF1 knockdown?)
C. Introduction:
79: “MDK correlated with poor prognosis in NSCLC [23]”- add here a sentence or more (2-3?) summarizing Hao et al 2013 PLoS One paper by the Japanese group which was the first to describe MDK expression in NSCLC, relevant signaling they studied, MDK inhibitor discovery, etcetera. Masui M et al Anticancer Res 2016 paper from the same group could also be added/ referenced.
D. Materials and methods.
- 91: “Dalls, TX’- change to “Dallas, TX”
- 127: collagen: supplier, catalog number?
- 202: HRO kit (Burlingame, CA)- add “Vector Laboratories”
- 212: NCCRI- in Seoul, South Korea? Please add.
E.Results:
- 253: in the text “”we noticed that” – please replace with “we evaluated”
- 262-263: “..and brain cancer cells were dependent on HIF-1 levels, and HCC cells were affected by Hif-2” – please change to “”..cell proliferation was dependent on Hif-1, and HCC cells proliferation was dependent on HIF-2 (under hypoxia).
- OK, after initial screen, you did go into studying A549 and H1299. Please again state the reason why you picked NSCLC cell lines and not breast, brain, etc. Also, if you go to original Japanese paper, Hao et al 2013 PLoS One, A549 is called “negative for MDK”, since the levels of MDK are very low relative to H441 and H520 cells. Why is there discrepancy? Please get H441, H520 cells, then run Western blot comparing relative MDK levels among H441, H520, A549 and H1299, relative to loading control, under normoxia vs hypoxia?
- 301-302: “three stromal cells were abrogated”- was this significant? Statistics? P-values?
- 375-376: “we treated with MDK inhibitor, iMDK, to measure the possibility of therapeutic target”- change to “MDK being a therapeutic target”. Also, After this sentence, please add a statement that prior Japanese data (Hao et al, 2013 PLoS One 2013 –[please reformat REF 30]) - have already shown in a different lung cell line iMDK efficacy.
F. Figures/ Figure legends:
Figure 1: change title to “Cell lines growth was more dependent on HIF-1 than HIF2” – not just the lung, but you can say the same about breast, gastric, brain and ovarian… Another major question is: you say that you measured cell growth for 16 hours only. How is it even possible to get 60% difference?? This is hard to believe, and please provide detailed explanation about this. In 16 hrs, I would only expect at most 5% difference.
Figure 2. Please add a Western blot comparing baseline MDK (and HIF1) relative protein expression vs actin / loading for H441, H520 (based on Japanese paper, these have very high MDK levels), then A549, H1299. Please correct corresponding Results text.
Figure 3. Add to the title “in NSCLC cells”. B- “MDK overexpression” – please add. A/B vs C/D – is it the same experiment? If so, could you clarify it? E: could you provide representative pictures?? And, pertinent to all Fig 3: where is the data / Westerns confirming MDK overexpression (best to show both Western and QPCR) – in A549 and H1299 cells –please add it. “movement of HUVEC cells was significant” – what was the stats? P values?
Figure 4: Please add the word “MDK related signaling” to the title, since you check signaling. Since the Japanese papr [corrected/ formatted REF 30] did check some apoptosis, you should cite it in corresponding Results text, and also, since they showed VEGF-A regulation, please also check in at least A549 and H1299 cells, whether VEGF-A was expressed / regulated by MDK (A549 for sure should express VEGF-A).
Figure 5: Please add either Western or IHC for: XIAP/ E-cadherin and/or vimentin, +/- VEGf-A, +/- casp-3 and Ki-67 for xenograft data.
Figure 6 and Figure 7: why there was no con + MDKi group? You are missing an important control here, since the drug may have effects in control group (which could be unrelated to MDK expression at all). This is certainly a deficiency.
G.Discussion:
- 466: …”..levels of oxygen stabilize HIFs protein and activates”- change to “stabilize HIF proteins and activate..”
- 472-473: “interest..in distinguishing the roles of HIF-1 and HIF2.”- unfinished sentence, in what process? (or in “distinguishing the distinct biological roles of..” ??
- 485: “especially, H1299 and A549 cells have an effect on..” – did you mean that “depletion of HIF1 had effect on proliferation in H1299 and A549 cells, while HIF2 depletion did not?? - please clarify.
- 488: “..prognostic factor in NSCLC”- could you add specific REF?
- 492: “..we profiled the top 14 cytokines secreted HIF1 induced NSCLC”- hypoxia-induced? Or NSCLC cells with induced HIF1? Also, Please add here as well a sentence explaining why did you specifically chose/ pick NSCLC and not breast, ovarian, brain cells for your study?? Rationale?
- 498: “..increased surving and XIAP”-change to “survivin and XIAP”
- 499-500: ..pro-apoptotic proteins (Fig 4E)- add “consistent with previously published NSCLC data with MDK, Hao et al – (please cite reformatted REF 30).
H. REFERENCES
All REFs have duplicated numbers
REF 30, please fix: first names instead of last names in that REF are currently listed
Author Response
Review 4
Comments and Suggestions for Authors
Overall, the original research paper entitled “Midkine is a potential therapeutic target of tumorigenesis, angiogenesis and metastasis in NSCLC” is of some interest to the readers of Cancers, even though does not add much novelty compared to previously published data. It has several deficiencies, and requires major revisions.
- As a major caveat, the major novelty is regulation of MDK by HIF1, as well as NF-kB/ Notch/EMT signaling data. The role of MDK in NSCLC, development of iMDK inhibitor to MDK (including xenograft studies), and even signaling / apoptosis regulated by MDK in NSCLC were all previously published by the Japanese group (Hao et al, Naomoto - PLoS One 2013 – which is actually cited by you in REF 30- with an incorrect format (first names instead of last names in that REF) and in another paper by the same group, Masui M et al Anticancer Res 2016).
Thank you for your comments, I fully understand your concerns that novelty or originality of my manuscript is low. However, if we could review thoroughly my manuscript, we will find that this manuscript gets further study from Hao et al. and Masui et al. First of all, the biggest finding is that Hao et al. screened MDK inhibitor and found iMDK which reduced endogenous expression of MDK. They performed anti-cancer study of iMDK with MDK-positive H441 and H520 lung adenocarcinoma cells in vitro and tumor growth in a xenograft mouse model in vivo. Masui et al. analyzed the antitumor effect of iMDK against HSC-2 and SAS cells oral squamous cell carcinoma. iMDK inhibited the proliferation of these cells dose-dependent via inhibiting phospho-ERK. They performed in vivo experiment with two groups of control and iMDK. Although iMDK decreased lung cancer volume, they did not have MDK expressed tumor growth data. However, we performed in vivo experiment with three groups of control, MDK expressed, and MDK expressed+iMDK. The tumor volume and weight of MDK expressed group was much bigger than control and the volume and weight of MDK+iMDK group was much lower than even in control. We suggest that MDK is very important of lung cancer progression and also iMDK might be a potential therapeutic target of MDK expressed lung cancer in vivo model. In addition, we found the margin of MDK expressed tissue in xenograft model was very irregular and increased angiogenesis compared with control. To confirm whether MDK increases lung cancer metastasis or not, we performed tumor xenografts in mice and spontaneous lung metastasis model and lung orthotopic model. We proved MDK increased lung cancer metastasis and iMDK decreased metastasis compared with even in control group. We also approved MDK increased angiogenesis through promoting endothelial migration, increased migration via Notch-NF-kB-Hes1 signaling in vitro and increased metastasis in two in vivo models. Second, your another concern was that we did cite incorrect format of Hao et al. We did correctly cite Hao et al. study as you mentioned.
B.Abstract:
“Cells were incubated under hypoxic conditions”: change to “Cancer cells were incubated…”
Thank you for your comment, I changed that as you mentioned and I wrote that in line 16.
“…the viability ..showed more sensitivity to HIF-01 that HIF-2”- could you rephrase/ perhaps say that “cell proliferation of XXX cells (? Under hypoxia) was more dependent on HIF1, except HCC cells where it was more dependent on HIF2”?
Thank you for your comment, I rephrased this sentence to cell proliferation of NSCLC, breast, gastric, and brain cancer cells under hypoxia status was more dependent on HIF-1a, except HCC cells where it was more dependent on HIF-2a.
I would add a few words on why NSCLC were chosen for further study and not other cell types (more suppression of proliferation after HIF1 knockdown?)
Thank you for your detailed revision, we mentioned why we did choose NSCLCs, especially H1299 for cytokine array from line 254 to 255. In addition, we describe H1299 cell line used in Figure 1D in line 255.
- Introduction:
79: “MDK correlated with poor prognosis in NSCLC [23]”- add here a sentence or more (2-3?) summarizing Hao et al 2013 PLoS One paper by the Japanese group which was the first to describe MDK expression in NSCLC, relevant signaling they studied, MDK inhibitor discovery, etcetera. Masui M et al Anticancer Res 2016 paper from the same group could also be added/ referenced.
Thank you for your comments, we summarized the study of Hao at al. and Masui M et al. and also cited two studies as a correct format from line 71 to line 74.
- Materials and methods.
- 91: “Dalls, TX’- change to “Dallas, TX”
We changed Dalls to Dallas
- 127: collagen: supplier, catalog number?
We added collagen information, (Merck Millipore (08-115), Frankfurter, Germany)
- 202: HRO kit (Burlingame, CA)- add “Vector Laboratories”
We added like that Vector Laboratories, Burlingame, CA
- 212: NCCRI- in Seoul, South Korea? Please add.
We changed that NCCRI, Goyang-si, South Korea.
E.Results:
- 253: in the text “”we noticed that” – please replace with “we evaluated”
We changed noticed to evaluated as you mentioned in line 241.
- 262-263: “..and brain cancer cells were dependent on HIF-1 levels, and HCC cells were affected by Hif-2” – please change to “”..cell proliferation was dependent on Hif-1, and HCC cells proliferation was dependent on HIF-2 (under hypoxia).
We changed that sentence as you mentioned. Cell proliferation of H1299, A549 (NSCLC), MCF7, MDAMB231 (Breast cancer), MKN1, MKN45 (Gastric cancer), and U87MG, SHSY5Y (Brain cancer) cells was dependent on HIF-1a, and HCC cells proliferation was dependent on HIF-2a under hypoxia (Figure 1A).
- OK, after initial screen, you did go into studying A549 and H1299. Please again state the reason why you picked NSCLC cell lines and not breast, brain, etc. Also, if you go to original Japanese paper, Hao et al 2013 PLoS One, A549 is called “negative for MDK”, since the levels of MDK are very low relative to H441 and H520 cells. Why is there discrepancy? Please get H441, H520 cells, then run Western blot comparing relative MDK levels among H441, H520, A549 and H1299, relative to loading control, under normoxia vs hypoxia?
MDK protein and mRNA levels of A549 cell were much less than that in H441 or H520 cells in normoxia like Hao et al. PLos One, 2013. However, MDK protein and mRNA levels of A549 cell were not much less than that in H441 or H520 in hypoxia condition. Because HIF-1a enhanced the transcription of MDK, acting on HIF-1a regulatory elements located in the MDK gene promoter (Paul et al. JBC 2004). In addition, we checked cell line authentication of H1299 and A549 through short tandem repeat genotyping. We discussed this issue from line 264 and to 270.
- 301-302: “three stromal cells were abrogated”- was this significant? Statistics? P-values?
Thank you for your comments, we described significant, statistics, and P-values of Figure 3B in line from line 326 to line 327.
- 375-376: “we treated with MDK inhibitor, iMDK, to measure the possibility of therapeutic target”- change to “MDK being a therapeutic target”. Also, After this sentence, please add a statement that prior Japanese data (Hao et al, 2013 PLoS One 2013 –[please reformat REF 30]) - have already shown in a different lung cell line iMDK efficacy.
Thank you for your comment, we described that Hao et al. reported that iMDK suppressed endogenous MDK expression in lung adenocarcinoma cells and reduced tumor volume derived from H441 lung adenocarcinoma cells in a xenograft mouse model from line 360 to line 361.
- Figures/ Figure legends:
Figure 1: change title to “Cell lines growth was more dependent on HIF-1 than HIF2” – not just the lung, but you can say the same about breast, gastric, brain and ovarian…
Thank you for your comments, we changed that cell lines were more dependent on HIF-1a or HIF-2a in figure 1 title.
Another major question is: you say that you measured cell growth for 16 hours only. How is it even possible to get 60% difference?? This is hard to believe, and please provide detailed explanation about this. In 16 hrs, I would only expect at most 5% difference.
Thank you for your comment, first of all we really apologize that we should describe detailed figure legend in figure 1A. Cell lines were transfected with siRNA for con, HIF-1a, and HIF-2a and then these cell lines were incubated for 16 h under normoxia and hypoxia. Cell viability was measured using MTS at 56 h after normoxia and hypoxia 16 h. Therefore, it had been taken total 72 h to measure cell viability after starting normoxia and hypoxia. We described more detail figure legend in line 281.
Figure 2. Please add a Western blot comparing baseline MDK (and HIF1) relative protein expression vs actin / loading for H441, H520 (based on Japanese paper, these have very high MDK levels), then A549, H1299. Please correct corresponding Results text.
Thank you for your comment, MDK protein and mRNA levels of A549 cell showed big difference compared with H441 or H520 cells in normoxia like Hao et al. PLos One, 2013. However, MDK protein and mRNA levels of A549 cell did not showed big difference compared with H441 or H520 in hypoxia condition as you mentioned in 7 question of results. We discussed this issue from line 266 and to 270.
Figure 3. Add to the title “in NSCLC cells”.
Thank you for your comments, we changed that MDK promoted endothelial cell migration and angiogenesis in NSLC cells line 316.
B- “MDK overexpression” – please add.
Thank you for your comments, we added MDK overexpression in Figure 3A.
A/B vs C/D – is it the same experiment? If so, could you clarify it?
We changed figure number to clarify, A and B to A, C and D to B. We also changed Figure B legend that relative numbers of cells transmigrated in Figure 3A in line 323.
E: could you provide representative pictures??
Thank you for your comments, we did provide representative pictures in Figure S5.
And, pertinent to all Fig 3: where is the data / Westerns confirming MDK overexpression (best to show both Western and QPCR) – in A549 and H1299 cells
Thank you for your comments, we performed western blotting with H1299 and A549 cells transfected with MDK plasmid. We described this result in line 297.
–please add it. “movement of HUVEC cells was significant” – what was the stats? P values?
Thanks for your comment, Thank you for your comments, we described significant, statistics, and P-values of Figure 3B in line from line 327 to line 333.
Figure 4: Please add the word “MDK related signaling” to the title, since you check signaling.
Thank you for your comments, we changed the title to MDK knock down inhibits MDK related signaling.
Since the Japanese papr [corrected/ formatted REF 30] did check some apoptosis, you should cite it in corresponding Results text,
Thank you for your comments, we cited that iMDK suppressed the PI3K and induced the apoptotic pathway in H441 cells in Hao et al. PLos One. 2013. We described that from line 360 to line 361.
and also, since they showed VEGF-A regulation, please also check in at least A549 and H1299 cells, whether VEGF-A was expressed / regulated by MDK (A549 for sure should express VEGF-A).
Thanks for your comments, hypoxia increased MDK and VEAF-A levels both H1299 and A549. However, hypoxia-induced VEGF-A did not decreased by MDK knock down under hypoxia status. These results suggest that VEGF-A expression did not dependent on MDK expression (Figure S???). We described these results from line 310 to line 312.
Figure 5: Please add either Western or IHC for: XIAP/ E-cadherin and/or vimentin, +/- VEGf-A, +/- casp-3 and Ki-67 for xenograft data.
Thank you for your comments, we performed Western blotting for XIAP, E-cadherin, Vimentin, VEGF-A, and cleaved-caspase 3 and IHC for Ki-67 with xenograft sample (Figure 5 and Figure S12). We described these results from line 402 to line 405.
Figure 6 and Figure 7: why there was no con + MDKi group? You are missing an important control here, since the drug may have effects in control group (which could be unrelated to MDK expression at all). This is certainly a deficiency.
Thanks for your comments, tumor tissue from xenograft or lung orthotopic model could be exposed by hypoxia. Even though hypoxia induced MDK, hypoxia-induced HIF-1a or HIF-2a could change many molecules. Therefore, we need to check MDK effects of pro-tumor and the anti-tumor effect of iMDK under MDK expression status. We described about this from line 376 to line 379.
G.Discussion:
- 466: …”..levels of oxygen stabilize HIFs protein and activates”- change to “stabilize HIF proteins and activate..”
We changed the diminished levels of oxygen stabilize HIF proteins and activate the HIFs-dependent transcription in line 488.
- 472-473: “interest..in distinguishing the roles of HIF-1 and HIF2.”- unfinished sentence, in what process? (or in “distinguishing the distinct biological roles of..” ??
Thank you for your comment, we finally decided to omit this sentence in line 492.
- 485: “especially, H1299 and A549 cells have an effect on..” – did you mean that “depletion of HIF1 had effect on proliferation in H1299 and A549 cells, while HIF2 depletion did not?? - please clarify.
Thank you for your comments, we changed that depletion of HIF-1a had the most effect on proliferation in H1299 and A549 cells in HIF-1a dependent cells compared with depletion of HIF-2a from line 506 to line 508.
- 488: “..prognostic factor in NSCLC”- could you add specific REF?
Thank you for your comments, we added specific reference Wang et al. Gene. 2014.
- 492: “..we profiled the top 14 cytokines secreted HIF1 induced NSCLC”- hypoxia-induced? Or NSCLC cells with induced HIF1? Also, Please add here as well a sentence explaining why did you specifically chose/ pick NSCLC and not breast, ovarian, brain cells for your study?? Rationale?
Thank you for your comment, we added why we choose H1299 cells. H1299 cell in NSCLC cells is the most affected on cell viability by HIF-1a knock-down from line 511 to line 512.
- 498: “..increased surving and XIAP”-change to “survivin and XIAP”
We changed surving to survivin.
- 499-500: ..pro-apoptotic proteins (Fig 4E)- add “consistent with previously published NSCLC data with MDK, Hao et al – (please cite reformatted REF 30).
Thank you for your comments, we added that as you mentioned above and cited Hao et al. PLosOne. 2013. in line 523.
- REFERENCES
All REFs have duplicated numbers
We changed all references had one number.
REF 30, please fix: first names instead of last names in that REF are currently listed
We fixed wrong first name of Hao et al.

Round 2
Reviewer 3 Report
This interesting manuscript has been revised thoroughly. My concerns have been addressed.
Reviewer 4 Report
OK, with added experiments and with revised version, the paper could be potentially accepted.
Several minor mainly English writing related comments / change requests (also, some that cite new data) are summarized below.
- Abstract: 20-22: rephrase this sentence to: “Among them, H1299 NSCLC cell line proliferation was the most affected by HIF1alpha knockdown.”
- Introduction: 71-72: rephrase this sentence to: “Hao et al discovered an MDK inhibitor, iMDK…. ” And 73-74: rephrase to: “..and SAS oral squamous carcinoma cells.”
- Materials and Methods: 121: “Frankfurter”- change to “Frankfurt”
- Results: 269: change “quite low” to “quite lower”. 342: change to “cell lines are affected by hypoxia-induced MDK.” 358: change “overexpressed surviving” to “overexpressed surviving”. 387: change to ”iMDK (9mg/kg) was intraperitoneallyinjected with..”. 405: change to: “…E-cadherin which were suppressed by iMDKtreatment (Figure S12). And insert on 405-406 “iMDK treatment also induced caspase-3 cleavage (Fig. S12).”.
- Discussion: 498-499: “…A549, (NSCLC)”, - here, insert H520, and H441; and 501 change to (S1, S2 and S8) under hypoxic conditions (OK, since Fig S8 does have H520, H441 HIF-1 induction expression under hypoxia).